# Age Matters: A Comparative Study of RF Heating of Epicardial and Endocardial Electronic Devices in Pediatric and Adult Phantoms during Cardiothoracic MRI

**DOI:** 10.3390/diagnostics13172847

**Published:** 2023-09-02

**Authors:** Fuchang Jiang, Kaylee R. Henry, Bhumi Bhusal, Pia Sanpitak, Gregory Webster, Andrada Popescu, Christina Laternser, Daniel Kim, Laleh Golestanirad

**Affiliations:** 1Department of Biomedical Engineering, Northwestern University, Evanston, IL 60208, USA; 2Department of Radiology, Northwestern University, Chicago, IL 60611, USA; 3Division of Cardiology, Ann and Robert H. Lurie Children’s Hospital of Chicago, Northwestern University, Chicago, IL 60611, USA; 4Division of Medical Imaging, Ann and Robert H. Lurie Children’s Hospital of Chicago, Northwestern University, Chicago, IL 60611, USA

**Keywords:** cardiac, epicardial leads, implants, imaging, MRI, pediatric, safety

## Abstract

This study focused on the potential risks of radiofrequency-induced heating of cardiac implantable electronic devices (CIEDs) in children and adults with epicardial and endocardial leads of varying lengths during cardiothoracic MRI scans. Infants and young children are the primary recipients of epicardial CIEDs, though the devices have not been approved as MR conditional by the FDA due to limited data, leading to pediatric hospitals either refusing the MRI service to most pediatric CIED patients or adopting a scan-all strategy based on results from adult studies. The study argues that risk–benefit decisions should be made on an individual basis. We used 120 clinically relevant epicardial and endocardial device configurations in adult and pediatric anthropomorphic phantoms to determine the temperature rise during RF exposure at 1.5 T. The results showed that there was significantly higher RF heating of epicardial leads than endocardial leads in the pediatric phantom, but not in the adult phantom. Additionally, body size and lead length significantly affected RF heating, with RF heating up to 12 °C observed in models based on younger children with short epicardial leads. The study provides evidence-based knowledge on RF-induced heating of CIEDs and highlights the importance of making individual risk–benefit decisions when assessing the potential risks of MRI scans in pediatric CIED patients.

## 1. Introduction

Cardiac implantable electronic devices (CIEDs) are a vital option to treat cardiac conduction disorders, replacing sinus and atrioventricular nodal function, and providing defibrillation for abnormal heart rhythms in both adults and children. Pacemakers, being the most prevalent among the various types of CIEDs, are utilized to restore and regulate the heart’s rhythm by delivering electrical impulses, thereby establishing a normal cardiac rhythm. Out of the global population of pacemaker recipients, an estimated 3 million individuals currently have pacemakers, and approximately 600,000 of these devices are implanted through surgery each year [1]. Among these, pediatric pacemaker implants account for approximately 1% of the total number of pacemaker implants performed [2]. Depending on the patient’s age or weight, there are two prevalent clinical approaches to pacemaker implantation. In adults and older children (>10–25 kg), the common practice for implanting CIEDs involves inserting leads through the subclavian vein and connecting them to an implantable pulse generator (IPG) placed in the subpectoral pocket [3]. Infants and young children, on the other hand, often receive an epicardial CIED system where the lead is sewn directly to the myocardium and the IPG is placed inferior to the abdominal rectus (See Figure 1). While these are the most common conventions in clinical practice, there have been cases where young children have received transvenous pacing systems and adult patients have received epicardial leads due to unusual venous anatomy [3].

Magnetic resonance imaging (MRI) is a powerful imaging modality with excellent soft-tissue contrast and a non-invasive nature that has become the preferred diagnostic imaging method for various neurological, cardiovascular, and musculoskeletal disorders. It is estimated that at least one MRI scan is necessary during the lifetime of 75% of patients with CIEDs, and many require repeated examination [4,5,6]. The demand for MRI scans is likely even higher in children, as developmental changes necessitate more frequent assessments and alternative imaging modalities with ionizing radiations are more restricted.

Unfortunately, the use of MRI is limited in patients with conductive medical implants due to risks associated with the interaction of MRI magnetic and electric fields with conductive devices. Specifically, an MRI machine produces three electromagnetic fields which can interfere with CIEDs: the static magnetic field (B_0_), the RF field (B_1_ and its associated E field), and the pulsed magnetic gradient field (i.e., Gx, Gy, Gz). The interaction with B_0_ is negligible in the modern era due to the low amount of ferromagnetic material in CIEDs. Gradient fields can induce electric currents in the device’s circuitry, potentially interfering with device function. The new generation of CIEDs, however, have enhanced programming and fail-safe MRI switches that minimize the interference. RF heating, on the other hand, remains a significant issue. This occurs due to the “antenna” effect, where the MRI transmit coil’s electric field couples with CIED leads and amplifies the energy deposition in the tissue at the tip of the lead, causing potential thermal injuries. To address this problem, comprehensive tests are performed by device manufacturers to determine safe conditions for MRI in patients with implants as reflected in the MR-conditional labeling of certain devices [7]. For endocardial CIEDs used in adults, MR-conditional labeling has been available since 2011. Many large studies have been conducted, allowing patients to safety undergo MRI exams at 1.5 T [8,9,10,11]. However, the FDA has never granted MR-conditional status for an epicardial device, and the limited data that is available indicates an elevated risk of radiofrequency (RF) heating of epicardial leads [12,13].

The Heart Rhythm Society’s 2017 expert consensus statement recommended that patients with CIEDs only undergo MRI when the product labeling is adhered to [13]. Despite this, the need for MRI has led some groups to scan patients with epicardial systems off label [14,15]. In response to this reality, the 2021 PACES guideline made a class IIb recommendation for CIED MRI patients with epicardial leads, based on individualized consideration of the risk/benefit ratio [16]. However, there is currently no guidance available on how to quantify these risks, and the decision to scan a patient and the choice of MRI protocol is based on the clinicians’ risks assessment, which is in turn based on limited literature.

The goal of this study was to compare MRI-induced RF heating around the tips of epicardial and endocardial leads, with different lengths and trajectories, inside adult and pediatric anthropomorphic phantoms during cardiothoracic MRI at 1.5 T. A comparative study of this nature is currently missing in the literature and is necessary to allow for evidence-based risk assessment. As body size and lead trajectory substantially affect RF heating [17,18,19,20,21,22,23], we performed experiments in human-shaped phantoms of different sizes, implanted with leads that followed patient-specific trajectories. To ensure that our results were robust and reproducible, we repeated experiments for a subset of cases and performed a test–retest reliability analysis. Finally, to provide a conservative framework that is relevant to clinical care, we performed experiments to determine imaging conditions that constrained RF heating to <3 °C (similar to a fever).

Our work provides knowledge on RF heating of both epicardial and endocardial leads in clinically relevant configurations as the basis for evidence-based assessment of risk-to-benefit ratio of performing MRI in children with CIEDs.

## 2. Materials and Methods

### 2.1. Phantom Design and Construction

Most studies that have assessed RF heating of conductive implants in the MRI environment have used a box-shaped phantom based on the ASTM standards [24]. However, a recent report highlighted a 10-fold discrepancy in the calculated specific absorption rate (SAR) in the ASTM box versus human body models [25]. This discrepancy was mainly attributed to the highly elliptical field polarization in the shallow-depth (~9 cm) ASTM phantom, which differs substantially from the field distribution in the human body. To address this issue, we developed custom-made human-shaped phantoms that mimicked the silhouette of an average-sized adult (length 65 cm, width 45 cm, depth 14 cm) and a twenty-nine-month-old child (length 53 cm, width 30 cm, depth 15 cm) (see Figure 2). The phantoms were filled with polyacrylamide (PAA) gel (22 L for the adult and 9 L for the pediatric phantom) with a conductivity of σ = 0.47 S/m and relative permittivity of $\varepsilon_{r}$ = 88, representing dielectric properties of a tissue-mimicking medium [7]. Care was taken to ensure that implanted leads and the IPG were submerged within the gel at a depth of 2–3 cm below the surface, mimicking the placement in patients.

### 2.2. Device Configurations

The magnitude, polarization, and phase distribution of MRI incident electric field along the length of an implant determine the RF-induced currents on elongated implants [26,27,28,29,30,31]. Consequently, the position and trajectory of an implanted lead has a significant impact on the RF heating around its tip [20,31,32,33]. Thus, a reliable and robust RF safety assessment requires patient-derived device configurations. We retrospectively reviewed chest X-ray and computed tomography (CT) images of 200 adult and pediatric patients with epicardial and endocardial CIEDs to create representative device trajectories. The Institutional Review Boards of Northwestern University and Lurie Children’s Hospital approved the use of patient imaging data for modeling and safety assessment purposes. To ensure reproducibility, we designed and 3D printed trajectory guides to route the leads along patient-specific trajectories and keep them securely in place during the experiments, similar to previous studies [34,35] (see Figure 2C). We conducted experiments on epicardial systems using three of the most commonly used lead lengths (i.e., 15 cm, 25 cm, and 35 cm) (Medtronic CapSure^®^ EPI 4965 (Minneapolice, MN, USA)). The leads were connected to a Medtronic Azure™ (Minneapolice, MN, USA) XT DR MRI SureScan pulse generator placed in the phantom’s abdomen. Experiments with the endocardial system used 35 cm, 45 cm, and 58 cm leads (CapSureFix Novus MRI™ SureScan™ bipolar 5076 (Minneapolice, MN, USA)) connected to the same pulse generator placed in the left infraclavicular region.

We performed 120 experiments in total, with 60 in each of the adult and child phantoms. We modeled clinically relevant trajectories using a combination of specific patient images, X-ray images available in the literature, and expert opinion, including input from author GW. We performed 60 experiments in the child phantom, with 15 cm and 25 cm epicardial leads and 35 cm and 45 cm endocardial leads routed along 15 patient-derived trajectories. Similarly, we conducted 60 experiments in the adult phantom, using 25 cm and 35 cm epicardial leads and 45 cm and 58 cm endocardial leads, each routed along 15 patient-derived trajectories. Figure 3 displays a few representative trajectories for each category. We can provide Computer-Aided Design (CAD) files for all 120 trajectories upon written request to the corresponding author. Both the epicardial pacing system and endocardial pacing system were powered off during the experiments.

### 2.3. Imaging Protocol

The temperature measurements were obtained using MR-compatible fiber optic probes (OSENSA, Vancouver, BC, Canada, resolution 0.01 °C) that were secured at the tip of the lead. A custom-made holder was 3D printed to ensure reliable thermal contact, allowing the temperature probe to remain in direct contact with the lead tip throughout the experiment. The RF exposure was conducted using a 1.5 T Siemens Aera scanner (Siemens Healthineers, Erlangen, Germany), with the phantoms positioned for cardiothoracic imaging. A high-SAR T1-weighted turbo spin echo (T1-TSE) sequence (TE = 7.3 ms, TR = 897 ms, B_1_^+^ = 5 µT, acquisition time = 280 s) was used. Although the acquisition time was shorter than typically used in RF heating experiments (i.e., ~5 min instead of 10 min), it was long enough to ensure that all temperature profiles reached a plateau, thereby ensuring the accuracy of our comparative temperature rise study.

### 2.4. Test-Retest Analysis

To assess the reliability of our measurements, we performed re-tests for 20% of the total number of cases (i.e., 24 experiments) and calculated the intraclass correlation coefficient (ICC). For each category, we selected trajectories that produced high, average, and low RF heating to maximize the heterogeneity of the re-test sample. ICC estimates and their 95% confidence intervals were calculated using RStudio (version 4.1.1) based on a single rating, absolute-agreement, two-way mixed-effects model [36].

### 2.5. Establishing Safe Exposure Limits Based on Worst-Case Scenarios

After conducting and validating comparative experiments, we identified device configurations that generated the highest heating for each category. Subsequently, we performed secondary experiments to determine the maximum RMS B_1_^+^ that generated less than 3 °C RF heating after 10 min of scanning. To achieve this, we implanted the worst-case RF heating device configurations into the phantoms and recorded the temperature at the lead’s tip during scanning with varying RMS B_1_^+^ values. To have complete control over the RF exposure characteristics, we disabled the gradient coils and transmitted a train of 1 ms rectangular RF pulses using the “rf_pulse” sequence from the Siemens Service Sequence directory. The flip angle was then adjusted to generate RMS B_1_^+^ values ranging from 2 µT to 5 µT. Pulse sequence parameters are summarized in Table 1 for each experiment. The maximum RMS B_1_^+^ allowed before the scanner stopped the operation was 5 µT, which corresponded to the scanner-reported SAR of 100%.

### 2.6. Statistical Analysis

A means comparison was implemented using t-tests to determine differences in heating between epicardial and endocardial leads in the adult and child phantoms. All statistical tests were run using RStudio (version 4.1.1) with a 95% confidence level.

## 3. Results

### 3.1. Reliability of Measurements

The test–retest experiments showed excellent reliability with an ICC of 0.95–0.99 at a 95% confidence interval. Appendix A provide the RF heating measurement values and corresponding device configurations.

### 3.2. RF Heating of CIEDs in the Pediatric Phantom

The mean ± SD RF heating in the pediatric phantom was 0.6 ± 0.4 °C for endocardial leads and 3.4 ± 3.0 °C for epicardial leads (pooled over both lead lengths in each case). The maximum RF heating recorded was ~12 °C for the 25 cm epicardial lead. For a 10-minute scan, this would be equivalent to a cumulative thermal dose of CEM43 °C = 640 min, high enough to cause necrosis in pig muscles [37].

The epicardial lead’s RF heating was significantly higher than that of endocardial leads in the pediatric phantom (*p* < 0.001). Within epicardial leads, the 25 cm lead generated higher RF heating than the 15 cm lead (4.4 ± 3.7 °C vs. 2.4 ± 1.5 °C, *p* < 0.05). Within endocardial leads, the 35 cm lead generated higher RF heating than the 45 cm lead (0.9 ± 0.4 °C vs. 0.4 ± 0.3 °C, *p* < 0.001).

### 3.3. RF Heating of CIEDs in the Adult Phantom

The mean ± SD RF heating in the adult phantom was 2.0 ± 1.8 °C for endocardial leads and 3.0 ± 3.2 °C for epicardial leads. A t-test comparison did not reveal a significant difference between RF heating of endocardial and epicardial leads (*p* = 0.16). The maximum RF heating recorded was ~12 °C for the 25 cm epicardial lead. Among epicardial leads, the 25 cm lead generated higher heating than the 35 cm lead (4.8 ± 3.5 °C vs. 1.2 ± 1.5 °C, *p* < 0.001). Among endocardial leads, the 45 cm lead generated higher heating than the 58 cm lead (3.2 ± 1.9 °C vs. 0.8 ± 0.7 °C, *p* < 0.001).

### 3.4. Adult vs. Pediatric Phantom

We found no significant difference between the RF heating of epicardial leads in the pediatric and adult phantoms (*p* = 0.65). However, endocardial leads in the pediatric phantom generated significantly less RF heating compared with endocardial leads in the adult phantom (0.6 ± 0.4 °C vs. 2.0 ± 1.8 °C, *p* < 0.001). This difference could be attributed to two factors. First, the incident MRI electric field tends to be lower in a child’s body compared with an adult body due to their smaller size and more central position within the RF coil. This can be observed in Figure 4. Figure 4 shows the simulated MRI incident E field (i.e., electric field in the body in the implant’s absence) in a 29-month-old body model compared with an adult body model, subject to RF exposure in a 64 MHz body coil. The input power of the coil was adjusted to generate a mean B_1_^+^ = 5 μT at the isocenter for both. CIED devices would be exposed to a higher incident electric field in the adult body compared with the child body (details of simulation setup and results are provided in the Appendix A).

The second factor is attributable to differences in lead trajectories related to children’s anatomy. Because the IPG is much closer to the heart in pediatric patients compared with adults, there will be more excess lead length, which is usually looped around the IPG. The addition of these loops could have contributed to the lower RF heating similar to what is observed in neuromodulation devices [38,39]. This can be specifically appreciated comparing RF heating of the 45 cm endocardial leads in adult (3.2 ± 1.9 °C) vs. child (0.4 ± 0.3 °C) phantoms, where the difference in RF heating can be solely attributed to differences in position and trajectory of the lead (Figure 5). Individual temperature rise for each case is provided in the Appendix A.

### 3.5. Effect of Lead’s Length

The maximum RF heating in both phantoms was observed for the 25 cm epicardial lead. The half wavelength of the radiofrequency field in the PAA ($\varepsilon_{r}$ = 88) was ~ 24.5 cm, which explains the high heating as the length of the lead approached the resonance length. This phenomenon, commonly referred to as the resonance effect, has also been observed in other types of wire implants [40,41].

### 3.6. Estimation of Safe RMS B_1_^+^ Levels Based on Worst-Case Observations

Temperature rise distribution histograms are provided in the Appendix A. For each case, the device configuration that generated the highest RF heating was used in subsequent experiments to estimate RMS B_1_^+^ values that constrained RF heating to <3 °C. Our data indicates that this approach was conservative, as >99% of cases in the general population would generate RF heating below this limit.

Figure 6 gives the RMS B_1_^+^ values and their corresponding RF heating for device configurations that generated highest heating in each category. For the pediatric phantom, both 35 cm and 45 cm endocardial leads generated RF heating well below 3 °C after 10 min of scanning at the maximum allowable RMS B_1_^+^ (i.e., 5µT) and, thus, are not shown in the graph. The maximum RMS B_1_^+^ that generated RF heating <3 °C was 2.5 µT for the 25 cm epicardial lead and 3.6 µT for the 15 cm epicardial lead.

For the adult phantom, the 58 cm endocardial lead generated RF heating well below 3 °C, so it is not shown in the graph. The maximum RMS B_1_^+^ that generated RF heating <3 °C was 2.3 µT and 3.2 µT for the 25 cm and 35 cm epicardial lead, respectively, and 3.3 µT for the 45 cm endocardial lead.

## 4. Discussion

### 4.1. Risk of MRI-Induced RF Heating in Patients with CIEDs

The severity of RF heating of CIEDs during MRI is influenced by two primary factors: the patient’s body size and the lead’s length/trajectory. The patient’s body size is a critical determinant because the distribution of the MRI electric field inside a sample is highly sensitive to its size and material composition. In children, the ratio of bone, fat, and muscle differs from that of adults, resulting in a substantially different electric field distribution at the CIED’s location. Therefore, safety assessments for MRI in children must be conducted using age-specific pediatric body models. The lead’s trajectory is the second determinant of RF heating and is correlated with the patient’s size, with smaller children receiving shorter leads. Even certain lead lengths that are common for both adults and children follow a different trajectory in children, requiring a different safety assessment. Our findings reveal that specific lead configurations in children generate significantly lower RF heating than the level deemed safe by the FDA for adults (e.g., 35 cm and 45 cm endocardial leads). However, some epicardial lead configurations generate excessive (e.g., 25 cm epicardial lead) RF heating which can cause myocardial lesions, underscoring the need for size- and lead-specific risk assessments.

There is a limited amount of available data concerning the MRI-induced RF heating of epicardial leads. In a recent in vitro study [12], temperature increase was measured at the tip of a commercially available bipolar epicardial lead (CapSureEpi 4968 with 35 cm, Medtronic) positioned in an ASTM phantom, with the lead location placed closer to the edge of the phantom. The trajectories included followed a straight path or coiling of varying lengths. However, in a realistic scenario, leads of pacemakers are usually looped at different locations to accommodate for the excess length. In this study, we specifically investigated the RF heating with leads along different realistic human-shape phantoms and aimed at obtaining a range of temperature changes to better represent the real-case scenario.

### 4.2. Rise of off-Label Scanning Is a Reality and Caution Is Warranted

So far, off-label MRI of CIED patients has been performed in several thousand adults and reported no significant side effects. The word “significant”, however, should be interpreted with caution. A recent study on adults with abandoned or functioning epicardial leads [42] found a transient elevation of pacing threshold in one patient, and an irreversible atrial pacing lead impedance elevation (>10,000 ohms) in another. While these may have been coincidental to the timing of the MRI, the retrospective nature of the study and the timing of the impedance rise make RF heating in the MRI scanner impossible to exclude. It is possible that RF heating in the MRI scanner led to a formation of low-conductivity scar tissue around the tip of the lead. Another study in pediatric and adult patients with epicardial or abandoned leads [14] reported a heat sensation in the left lateral chest at the location of an abandoned endocardial lead in one patient (36 years old)—which required the scan to be stopped—and a tingling sensation at the location of the epicardial lead in another. Even if these risks may be manageable in adults, there are additional factors that complicate accurate risk assessment in children. An infant or a 5 year old scanned under anesthesia cannot complain of a heating sensation to stop the MRI. The results could be an irreversible thermal injury of the heart tissue. Because myocardial scarring during early childhood can expand as the heart grows over time [43], such damages could cause long-term complications over the individual’s life span. While we did not find a systematic difference in RF heating of epicardial leads in pediatric vs. adult phantoms, marked heating elevations were present in clinically relevant scenarios. For example, our results show that MR-induced heating of epicardial leads reaches levels that are 267% higher in young children who mostly receive short epicardial leads (e.g., 25 cm) compared with adults who receive longer leads (e.g., 35 cm).

### 4.3. Perfusion in Accessing the Risk of MRI-Induced RF Heating

The wealth of national and international standards developed over the past two decades to ensure MRI safety unanimously recommend the use of perfusion-less phantoms to assess RF heating. The rationale behind this consensus is that neglecting cooling effects of perfusion which slightly overestimates RF heating adds a safety margin which is welcome when patient safety is at stake in a large population. The overestimation seems to be a reasonable trade off, considering that the epicardium, where the leads are affixed to, is excluded from circulation, and that in endocardium, which is cooled by circulation, the error due to exclusion of perfusion remains below 30% [13].

### 4.4. Limitations and Future Work

This work has several limitations. First, only a few epicardial and endocardial lead models were investigated. We are aware that other lead models exist in the market. Even within device models in our study, we limited our work to lead lengths that were most commonly used at our institution. For example, Medtronic 4965 epicardial leads also come in 50 cm length, and Medtronic 5076 endocardial leads also come in 52 cm, 65 cm and 85 cm lengths. Bipolar epicardial leads (e.g., Medtronic 4968 CapSure Epi (Minneapolice, MN, USA)) for biventricular or dual-chamber pacing were not investigated either. This was a compromise that allowed us to study each lead more thoroughly (i.e., more patient-derived trajectories) which we deemed to be crucial to infer conclusions on safety.

Second, this work only studied intact CIED devices; abandoned leads were not included. Abandoned leads are observed in a sizeable fraction of CIED patients for whom leads fail for technical reasons, infection, or blockage of the vein by clot. In such situations, the patient is referred for lead extraction and implantation of new leads [44] or lead abandonment and implantation of additional leads [45]. Extraction is not always successful in transvenous leads, and the risks of sternotomy and surgical lead extraction in patients with epicardial leads typically outweigh the benefits. This suggests that fractured epicardial leads are nearly always abandoned in place. Since MR-conditional labeling of a CIED only applies to the intact device with leads implanted in their original configuration, patients with abandoned leads are contraindicated for MRI. A recent study showed that capped abandoned leads induce tissue heating that is on average 3.5 times higher than what is generated by a complete CIED system [46]. Thus, the results of this study should not be generalized to cases of abandoned leads.

## Figures and Tables

**Figure 1 diagnostics-13-02847-f001:**
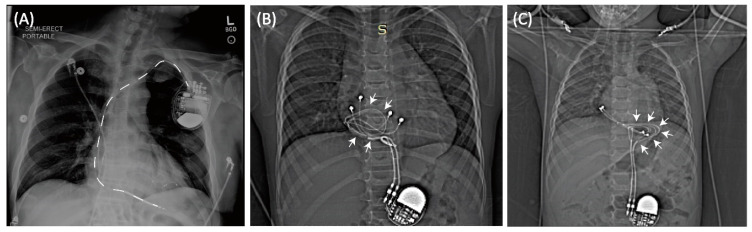
(**A**) Typical trajectory of an endocardial lead. Leads pass through the subclavian vein creating a stereotypical pathway in nearly all patients. In contrast, epicardial devices can have highly varied lead trajectories (**B**,**C**).

**Figure 2 diagnostics-13-02847-f002:**
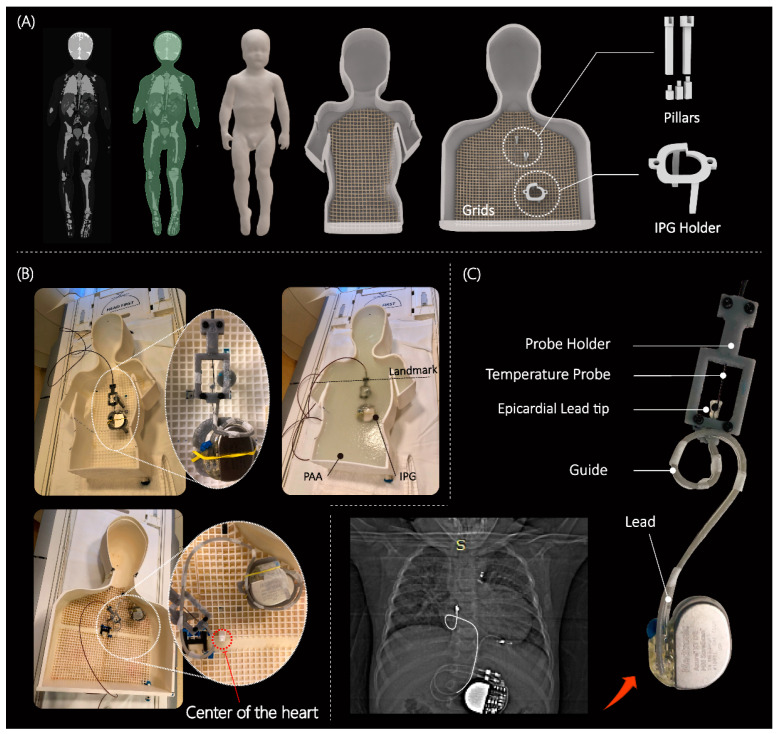
(**A**) 3D models of human-shaped pediatric and adult phantoms created from MRI of a 29-month-old child and representative images of an average-sized adult; the phantoms include grids, pillars, and different heights of pillar extensions to adjust the positioning of the CIEDs. (**B**) Assembled phantoms with implanted CIED and temperature measuring setup; cardiothoracic imaging landmark was used during MR scans. (**C**) An example of a patient-derived trajectory of an implanted lead obtained from X-ray image; close view of contact between the temperature probe and epicardial lead tip. The pediatric and adult phantom were 3D printed using Rigid 4000 resin (Formlabs, Somerville, MA, USA) and ABS Plastic, respectively. Pillars, holders, and extensions were made of Tough 1500 resin (Formlabs, Somerville, MA, USA).

**Figure 3 diagnostics-13-02847-f003:**
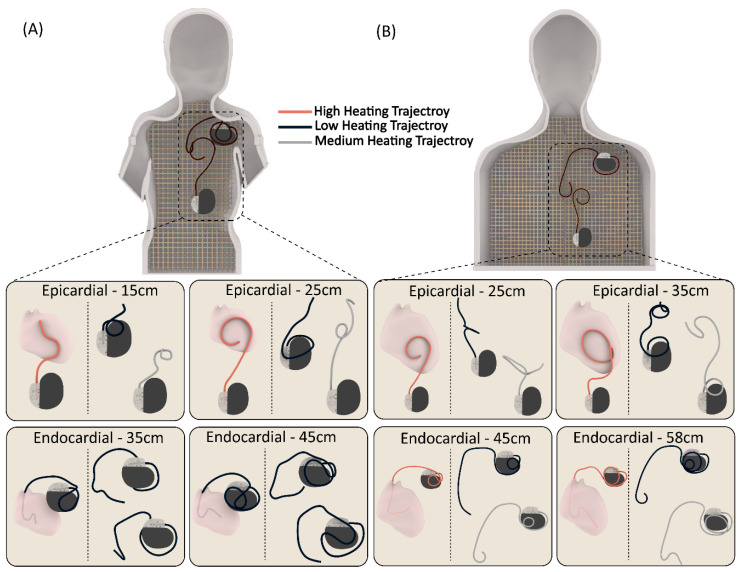
Representative patient-derived trajectories for each CIED category (24 trajectories in total) in (**A**) pediatric and (**B**) adult phantoms.

**Figure 4 diagnostics-13-02847-f004:**
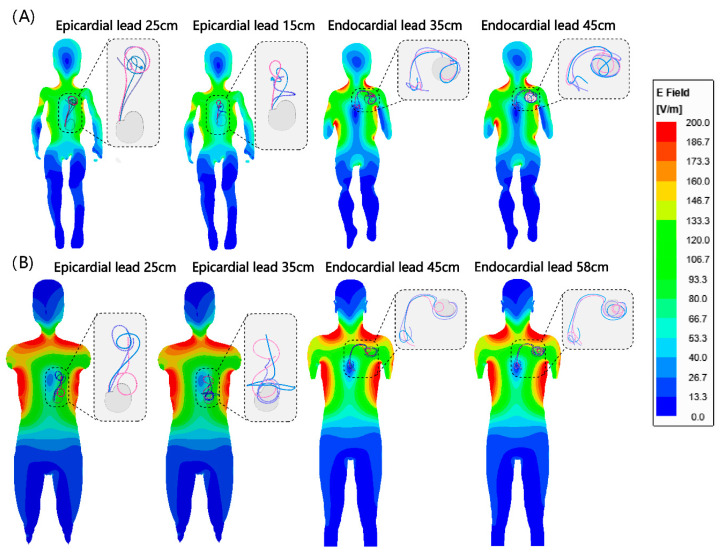
Simulated incident E field in (**A**) homogeneous child and (**B**) adult body models. Note that the incident field is the electric field of the MRI scanner in the absence of any implant. Trajectories of endocardial and epicardial devices are overlaid on the field maps for comparison. Details of simulation setting are provided in the Appendix A.

**Figure 5 diagnostics-13-02847-f005:**
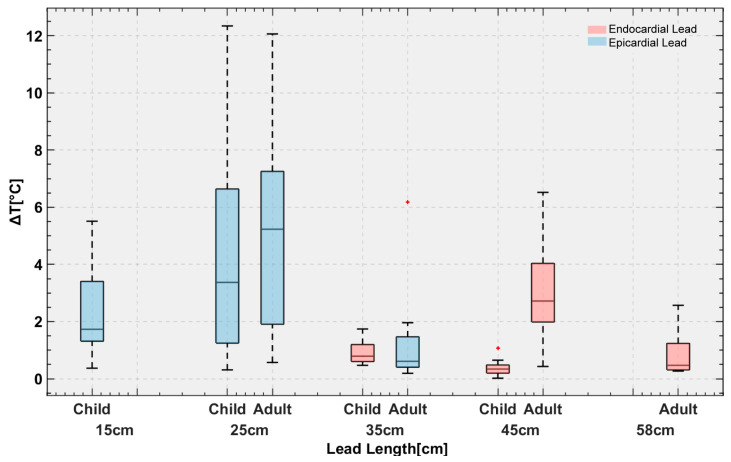
Temperature rise in the gel around the tips of the leads with 120 distinct trajectories implanted in pediatric and adult phantoms.

**Figure 6 diagnostics-13-02847-f006:**
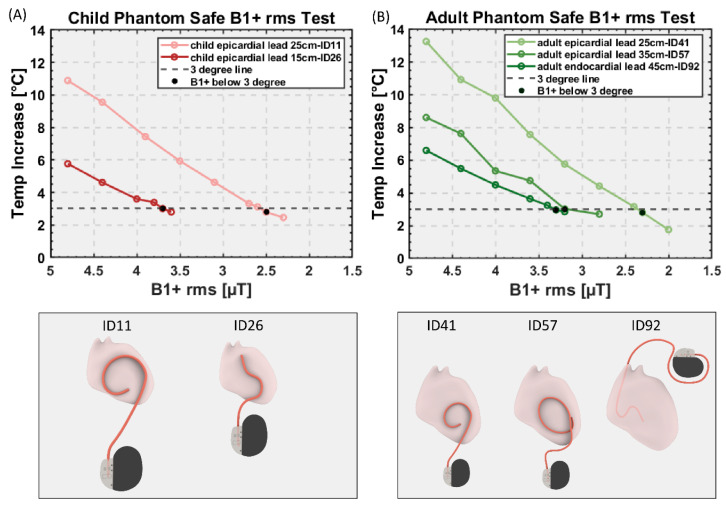
RMS B_1_^+^ values and their corresponding RF heating for device configurations that generated highest heating in each category for (**A**) child and (**B**) adult phantom. Black dots: RMS B_1_^+^ value that generates the temperature rise below 3 °C.

**Table 1 diagnostics-13-02847-t001:** RF pulse parameters to establish the safe exposure limits.

Frequency	Duration		TR
64 MHz	603 s		24.03 ms
Patient weight entered		Patient weight entered	
68.0 kg		13.6 kg	
Flip Angle (°)	B1 + rms [µT]	Flip Angle (°)	B1 + rms [µT]
360	4.80	360	4.82
330	4.40	330	4.41
300	4.00	295	3.95
270	3.60	285	3.81
240	3.20	278	3.72
210	2.80	270	3.61
180	2.40	265	3.55
170	2.27	245	3.28
150	2.00	235	3.15
		225	3.01
		205	2.74
		195	2.61
		185	2.48
		175	2.34

## Data Availability

Not applicable.

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
