# Peer review of "Age Matters: A Comparative Study of RF Heating of Epicardial and Endocardial Electronic Devices in Pediatric and Adult Phantoms during Cardiothoracic MRI"

_diagnostics, 2023, doi:10.3390/diagnostics13172847_

Round 1

Reviewer 1 Report

REVIEW 1: Age Matters: A Comparative Study of RF-Heating of Epicardial and Endocardial Electronic Devices in Pediatric and Adult Phantoms during Cardiothoracic MRI

Dear authors,

This manuscript focused on the potential risks of radiofrequency-induced heating of cardiac 16 implantable electronic devices. The results are interesting to assess the use of MRI in children with these devices.

The manuscript is interesting but I need you to answer some questions:

INTRODUCTION

The introduction is very short. Authors should include some epidemiological data to support their claims.

The introduction should not be to have images. Images should be attached as annexes.

MATERIALS AND METHODS

The authors must specify the research design.

DISCUSSION

The "Discussion" should not have sub-sections. The text should have a continuous narrative thread.

The information is redundant whit "Results". In the “Discussion”, the “Results” should be compared with other research.

The authors have not used a bibliography. The results need to be well explained with references to other studies.

CONCLUSIONS

There are no final "Conclusions".

REFERENCES

Many bibliographies are obsolete. The bibliographic citations used are more than 5 years old (42.5 %). The authors must update and arrange the bibliography.

Some references are incomplete or have errors. The authors should review this section.

Reviewer 2 Report

The authors have designed and conducted a study to assess heating due to RF at lead wire tips of pacemaker and implantable cardioverter/defibrillators at different clinically relevant lead lengths and orientations. The results show certain lead orientation produces higher temperature rise than others. The presented results have potential clinical relevance for cardiologists to consider during such lead placement. The study is especially relevant for pediatric population where CIEDs are usually implanted in patients after surgical correction of heart defects and require assessment of cardiac function by occasional utilizing radiographic imaging like CT and MRI.

The authors have discussed some of the major limitations of the study appropriately. The manuscript is well written. The author should address a few issues that would improve the clinical impact and relevance of their work.

Major issues

1.       Line 18: “Most children receive an epicardial CIED”. Authors need to cite appropriate reference to back this claim or rewrite the statement. In clinical practice, Transvenous lead placements are preferred, though the majority of epicardial lead placements are in children. This is done in neonates where transvenous lead placement is difficult due to their small size. This is typically done for children with either congenital heart block or acquired heart block after congenital heart surgery at age < 3years.

2.       Authors should specify if the CIEDS were powered ON or OFF during the experiments. While ICDs can be turned OFF for imaging studies, pacemakers usually need to operate continuously. Furthermore, the pacing rates and threshold voltages are different for pediatric and adult patients. Authors should briefly discuss the potential influence of this factor in an appropriate section (maybe in limitations).

3.      Discuss results in context of lead positioning modes. The study included lead tips position for atrial and ventricular pacing. Did authors observe any significant differences amongst these modes.

4.       All orientations covered were single lead positioned either into the right atrium or right ventricle. Were biventricular or dual chamber pacing cases excluded? If yes, the authors should address this in the study limitation or for potential future studies.

5.       Another major limitation of the study is the static homogenous environment used. The authors should briefly discuss how the temperatures rise observed in their setup would be different in in vivo setting (in patients). Given the dynamic environment in vivo, the reported temperature rise will probably be significantly less.

Minor issues

1.       Line 75 Remove “and” from Heart and Rhythm Society.  

2.       Line 95: add "and endocardial"  after epicardial

Round 2

Reviewer 1 Report

Dear authors,

I can understand your concern with my review and your decision of not to take into account my comments. They were only intended to improve the quality of your paper.

In this sense, I hope you understand my decision to inhibit in the review of your paper.

I trust in the criterium of the other reviewers for you benefit.

Sincerely.